# Risk Factors of Early Adolescence in the Criminal Career of Polish Offenders in the Light of Life Course Theory

**DOI:** 10.3390/ijerph18126583

**Published:** 2021-06-18

**Authors:** Krzysztof Pękala, Andrzej Kacprzak, Anna Pękala-Wojciechowska, Piotr Chomczyński, Michał Olszewski, Michał Marczak, Remigiusz Kozłowski, Dariusz Timler, Łukasz Zakonnik, Kamila Sienkiewicz, Elżbieta Kozłowska, Paweł Rasmus

**Affiliations:** 1Department of Medical Psychology, Faculty of Health Sciences, Medical University of Lodz, 90-131 Lodz, Poland; krzysztof.pekala@umed.lodz.pl (K.P.); pawel.rasmus@umed.lodz.pl (P.R.); 2Department of Applied Sociology and Social Work, Faculty of Economics and Sociology, University of Lodz, 90-136 Lodz, Poland; andrzej.kacprzak@uni.lodz.pl; 3Department of Clinical Pharmacology, Medical University of Lodz, 90-153 Lodz, Poland; anna.pekala-wojciechowska@umed.lodz.pl; 4Department of Sociology of Organization and Management, Faculty of Economics and Sociology, University of Lodz, 90-136 Lodz, Poland; piotr.chomczynski@uni.lodz.pl; 5Department of Management and Logistics in Healthcare, Medical University of Lodz, 90-131 Lodz, Poland; michal.olszewski@stud.umed.lodz.pl (M.O.); michal.marczak@umed.lodz.pl (M.M.); kamila.sienkiewicz@stud.umed.lodz.pl (K.S.); 6Center of Security Technologies in Logistics, Faculty of Management, University of Lodz, 90-237 Lodz, Poland; remigiusz.kozlowski@wz.uni.lodz.pl; 7Department of Emergency Medicine and Disaster Medicine, Medical University of Lodz, 92-212 Lodz, Poland; 8Faculty of Economics and Sociology, University of Lodz, 90-214 Lodz, Poland; lukasz.zakonnik@uni.lodz.pl; 9Department of Microbiology and Experimental Immunology, Medical University of Lodz, 92-213 Lodz, Poland; elzbieta.kozlowska@umed.lodz.pl

**Keywords:** life course theory, criminal career, criminology, risk factors, adolescence

## Abstract

Life course theory (LCT) diagnoses childhood and adolescent factors that determine an individual’s involvement in crime in the future. Farrington lists eight key correlates identified by empirical analyses of criminal careers. In this paper, we seek to discuss the inconsistencies with LCT that we observed in our three empirical studies of the criminal careers of Polish offenders. During 12 years of qualitative research, we conducted direct observations and in-depth interviews in juvenile correction institutions (21) and prisons (8) across the country. We gained access to incarcerated (102) and released (30) juvenile offenders, as well as to incarcerated (68) and released (28) adult offenders. We also conducted in-depth interviews (92) with experts working with young and adult offenders. We similarly accessed some offenders’ criminal records and psychological opinions. Our study revealed the strong presence of family and neighborhood influences on early criminality. Contrary to LCT assumptions, state-dependent institutions (military, work, family) were not strong enough determinants of delinquency. Polish offenders generally experience criminal onset later than LCT-oriented criminologists indicate. Based on our data, we also agree with the thesis that the onset of crime should be discussed as different age-related periods rather than just a general onset.

## 1. Introduction

The dynamics of crime in individual biographies, which is at the core of the life course theory approach, has been a central focus of criminologists from the beginning of the discipline to the present day [1] (pp. 23–24). LCT focuses on three key processes that can be distinguished in a criminal career: initiation of criminal behavior, persistence in crime, and desistance from crime. In this paper, we turn our attention to the first stage of the criminal career: the environmental conditions that contribute to its initiation.

Early sociologically oriented analyses in criminology recognized the links between crime and social disadvantages, such as reduced opportunities for social advancement, poverty educational deficits [2], being raised in degraded neighborhoods [3], deficits in family relations [4], the difficult life situation of families of origin, or the intergenerational transmission of patterns of deviant behavior [5]. This link between crime and class affiliation has often been taken up in later years as well.

Gendreau, Little, and Goggin [6] (pp. 580–584) performed a meta-analysis of 131 recidivism studies conducted between 1970 and 1994 in the United States and Canada. On their basis, 18 groups of social, economic, and psychological factors increasing the risk of recidivism were identified. Symptomatically, a large proportion of the most important of these related to environmental determinants of individual childhood and adolescence, including family factors (rearing methods, family criminality, family structure), low social achievements (e.g., low education), substance abuse, and “criminogenic needs” (e.g., antisocial cognitions, values, and behaviors resulting from membership in criminogenic environments).

Juvenile delinquency is of particular interest to criminology and life course theory and, therefore, so are the risk factors for juvenile delinquency interest. There is widespread agreement that early criminal initiation is a strong predictor of a long and prolific criminal career [7] (pp. 53–54), [8,9,10]. Thus, there is a high level of social (and academic) sensitivity to early symptoms of “social maladjustment”, because such manifestations indicate a risk of amplifying adult crime rates in the future. Over the years, numerous studies (mostly longitudinal or using a biographical method) have addressed this topic, producing a long list of childhood and adolescent risk factors for an individual’s initiation and development of a criminal career [8,11,12,13]. The literature indicates that juvenile offenders disproportionately come from neglectful backgrounds with cumulative disadvantages in which the child’s basic needs are not met [14] (pp. 207–210), [15] (pp. 3–4).

Criminologists view family in terms of a milieu with preventive, anti-criminal potential or, conversely, criminogenic and antisocial potential [16] (pp. 64–71). It has been argued that strong family ties are followed by a high level of social control that can deter an individual from criminal activity [17] (pp. 224–225). On the other hand, Adverse Childhood Experiences (ACEs) are noteworthy in the initiation of criminal careers [8,14,18,19], and the inability of the family to provide social control may be a major reason for turning to peer groups [17]. Hirschi [20] (pp. 90–92), [21] (p. 99) noted that in criminal/delinquent families, parents often do not know where their children are, rarely explain rules of conduct to their children, and do not find time to talk about their children’s important issues. Patterson [22] (pp. 80–89) went even further, claiming that parents who fail to clearly state (house) rules, monitor the child’s attitudes, track what their child is doing, reinforce pro-conformist behaviors, and include children in solving conflicts and disagreements are the prime determining variables of their children’s future delinquency.

Criminology identifies a long list of potentially criminogenic characteristics that recur in early offender life histories. These include a lack of parental supervision, strained family ties (e.g., due to conflict, parental alcohol addiction, domestic violence, child abuse), a criminal family history, a broken family structure often followed by child neglect and social orphanhood, and being raised in foster care, all of which are significant risk factors in themselves.

One of the most important childhood risk factors highlighted by researchers is family intergenerational delinquency, which we can refer to as “vertical” delinquency. This issue has long been the subject of empirical analysis. One of the most notable longitudinal studies conducted to date is the Cambridge Study in Delinquent Development (CSDD). It included 411 boys born in 1953 in a working-class inner-city neighborhood [23] (p. 680). Starting from 1961–1962, respondents were contacted every 10 years, and the biographical lives of the cohort were reconstructed through questionnaire interviews. It was found that almost half (48%) of the boys whose parents had been in prison before the age of 10 had also been sentenced to prison (in a control group of boys from single-parent families where the parents had not been in prison, one in four respondents had served a sentence). When sibling criminality (horizontal criminality) was taken into account, this percentage increased to 63% [24]. Glaze and Maruschak [25] (p. 7), who studied the cases of over 18,000 prisoners, showed that one in two people in prison in the United States is closely related to another person with a criminal record.

The CSDD also showed that childhood risk factors may be related to family structure. According to the study, boys from large families were criminalized twice as often as their peers from control families [26] (p. 159). Furthermore, Brownflied and Sorenson (1994, as cited in Farrington [14] (p. 207) noted that there is a greater risk of criminal behavior of further born children who receive less attention from their parents.

As indicated by life cycle theorists, most criminal careers begin in childhood or early adolescence, between the ages of 8 and 14 [27] (pp. 251–252), reaching a peak of criminal activity between the ages of 17 and 18 [12] (p. 51). Bernasiewicz and Noszczyk-Bernasiewicz [28] argued that the onset of delinquency occurs between the ages of 12 and 17. There has also been consensus among criminologists [29] that offenders with an early onset of delinquency are often associated with a higher frequency of offending. However, for adult female offenders, age is revealed in a scatterplot and “does not appear to illustrate any regularity” [30] (p. 122).

Koppen [13] (p. 94) as well as Thornberry [31] advocated the idea that instead of discussing the general onset of criminal careers, we should look at “different factors explaining the onset of crime at different ages”. Koppen [13] (p. 94) argued that “important factors explaining the onset of delinquency in early childhood (before age six), for example, are neuropsychological deficits and poor parenting. Individuals who start committing crimes in later childhood (ages 6–12) are influenced by their family and neighborhood, while adolescent offenders (ages 12–18) are influenced by their peers.”

## 2. Materials and Methods

This paper gathered data from three different projects. The first and most recent of these [32] took place between 2018 and 2020 (Grant of the Polish Ministry of Justice, No. DFS-II-7211-169/18/18, titled: “Social determinants of juvenile and adult crime” (project coordinator: Piotr Chomczyński). The full report on this research project (in Polish) is available online [30]). A total of 130 individuals participated in the study, of which 90 were juvenile (30) and adult (60) offenders, of which 30 were recidivists. More than half of the offender population was serving sentences in prison (55), and the rest were already at large (35). Juvenile and adult offenders were convicted of various types of crimes (robbery, drug trafficking, theft, homicide, physical assault). Most (28) of the expert interviews (40) were conducted in both juvenile (correctional) and adult (correctional and detention) facilities. They worked as psychologists, social workers, rehabilitation specialists, professional counselors, educators, and prison staff.

The second project was based on a long-term organizational ethnography between 2008 and 2017. One of the authors (Piotr Chomczyński) spent a total of nine months in all the types of reformatories and juvenile detention centers for boys (17) and girls (4) across Poland. The research was based on open-ended interviews conducted with male (43) and female (29) inmates, aged 13 to 21 years. Using overt participant observation, the author participated in all activities performed by the inmates, assisted in workshops, vocational training, during meals, in residential cells, and in leisure time. In addition, files (26) and psychological opinions (182) of some inmates from 2002 to 2015 were made available. Both files and opinions were disclosed to the researcher in a haphazard and arbitrary manner by the management of some correctional institutions, so no generalizations could be made on this basis. However, the documents shed light on certain regularities in the study population.

The third project took place between 2011 and 2018 and was based on qualitative research methods. One of the authors (Andrzej Kacprzak) conducted the study with (28) former prisoners (recruited via NGOs and social welfare institutions) and (8) current prisoners in correctional facilities for adults across Poland. The author used an open-ended biographical interview with adult offenders. These were supplemented by semi-structured interviews (10) with persons working with ex- or current prisoners (social workers, psychotherapists, NGO workers). The study focused on social factors that were barriers to the social integration of former inmates returning to society, as well as factors that promoted diversion from crime.

In all three studies, we used open-ended biographical tools that allowed our interviewees to make spontaneous statements. [33,34,35,36]. Our initial experience showed that inmates expressed reluctance to discuss their criminal experiences when they saw the printed questionnaire, so we used open-ended questions, like an informal conversation. Initially, we thought this was an artefact of our outsider status. However, we found that their hesitation could be overcome by making our approach more interactive and participatory. We memorized categories and invited respondents to develop and co-create interview scenarios by adding their own questions or editing ours. This gave us a richer description than a formal interview tool, and our interviewees became less suspicious [18,35,36,37,38,39]. We believe that this method put our subjects on a more egalitarian footing [33,35,40,41].

The duration of the interviews ranged from 30 min to 2 h, depending on the environmental conditions and the degree of trust placed in each respondent. The median age of adult offenders was 29 years, whereas of juvenile ones, it was 16. The median age of initiation of criminal activity for adult offenders was 17 years, and for juveniles it was 11 years. The interviewers approached the topic of criminal careers carefully and made a conscious effort to create an atmosphere of openness and trust in order to obtain the most objective and detailed data possible. To facilitate this, they disclosed information about their personal biography and research objectives.

Triangulation of our data was particularly important, as we were outsiders and dealing with a sensitive subject [42,43].

We analyzed our data using ATLAS TI software (ATLAS.ti Scientific Software Development GmbH, Berlin, Germany). Key categories were inductively generated through open and selective coding of interviews and illustrated with quotes presented in the paper that most closely reflected the conceptual framework discussed here [44,45,46,47]. To ensure anonymity and confidentiality, which are particularly needed when dealing with sensitive topics, all names used in our research are pseudonyms [32,33]. We edited the length and content of the quotations, when necessary, to protect our subjects from being identified.

## 3. Results: Environment-Related Risk Factors in Polish Offenders’ Criminal Careers

### 3.1. Criminal Onset

Statistics of the Polish Ministry of Justice seemed to confirm [14] Farrington’s and MacLeod’s [12] findings and reveal that between 2010 and 2016, both girls and boys reached the peak of criminal behavior at ages 15 and 16 [48] (p. 18).

In most of the adult biographies we analyzed, criminal onset occurred at later stages, most often between the ages of 12 and 17 [28]; however, the juvenile offenders of both genders who participated in our study typically begun their criminal careers between the ages of 11 and 12. Our data confirmed that early start of delinquency is often associated with higher rates of offending [29]. Our in-depth interviews also signalized gender differences in the onset of delinquency among adult offenders, as women tended to commit crimes later than men. There was no single answer to the question of gender differences in the onset of delinquency in the literature, but educators who have worked with both male and female juvenile offenders have indicated that girls’ aggressive behaviors are less predictable and more emotional compared to boys. Of course, this does not explain the abnormalities in girls’ delinquency, but it does allow us to see some gender-related differences:

“I worked both with girls and boys for over 20 years. Girls are much more emotional and less predictable than boys that are straighter in acting and easily driven by environmental stimuli. Among boys, you know what’s going on, but with girls, conflicts and aggression comes from nowhere”.(Kasia, 45, educator in juvenile detention center)

Our biography-based in-depth interviews with both boys and psychologists working in juvenile detention centers did not confirm/reject the neuropsychological risk factors in early-childhood. However, psychologists/therapists agreed that early childhood trauma and poor parental experiences were strongly overrepresented.

“According to my files’ desk research, daily observations, and talks with boys, many of them were exposed to trauma when they were very young. Usually, their parents and relatives were responsible for this”.(psychologist, juvenile detention center)

Our data revealed that the older our respondents were, the more they were influenced by peers and less by relatives and neighbors [13,31]. As their autonomy and mobility increased, place-bound significant others (family and neighbors) were gradually replaced by friends as “people by choice”. Very often, “the replacement process” took the form of running away from home as juveniles explored the outside environment and new friendships. The stories of our respondents demonstrated that runaways rarely limited themselves to just one escape and began to define it as a response to trouble, which can be interpreted as a symptom of difficult relationships in their family environment [49,50].

R:Mhm. Tell me, have you ever run away from home?

I:Quite often.

R:Yes? But why?

I:Well, how my mother gave me a penalty, for example, right? […] At the beginning, I used to run away to meet my friends and so on, right? […] And then, it’s some melange, no?

R:Mhm. Melanges, yes? Well, can you say more about these escapes?

I:Well, I used to come home after a while, right? [then I] go to one friend and to another.

### 3.2. Family-Related Crime Risk

Our data allowed us to confirm family-related risk factors that that recur in offenders’ early life stories, such as parental alcohol dependence, domestic violence, and child abuse. In most cases, the narratives of our interviewees revealed a pattern in which fathers were incarcerated and did not participate in child rearing.

R:Was there anybody from your family ever punished?

I:Father was punished many times. […] For some break-ins, beatings. Such things.

Fathers were also responsible for nonconformism, transmission of criminal values, and internalization of prison subculture. Their criminal path was well known to family members, who incorporated the general attitude of the father criminalizing. In situations where fathers did not engage in domestic violence but directed their aggression outside the family, juvenile offenders tended to glorify their fathers as “tough guys”.

I:My mother raised me.

R:And what about your dad?

I:[…] I don’t know him much, because he spent a lot of time in prison … He beat a policeman severely, he got 15 years, he made me at an intimate visit… I was born… when I was six years old, he left. I met him, he was with me for four years, then he went to prison again… and then again and finally was released half a year ago.

In a very different way, juvenile offenders and recidivists perceived their fathers, who both conflicted with law and used domestic violence. As our subjects were forced to defend their mothers and/or younger siblings, they acquired the role of the “violent defender” who fights back and “becomes a man” for the first time [51]. As “privileged” defenders, they mentally mixed noble motives with the use of violence. In most cases, it was a highly emotional experience, based on the use of violence as a solution to a problem, giving rise to subsequent lawbreaking.

I:Holidays were usually heavily sprinkled with alcohol. I do not wish that to anyone. I was avoiding this house and being around. As a kid, you know that I had to, but if I didn’t have to … My father got hit in the head when I was 17 years old.

R:What happened then?

I:I stood up for my mother. I came back home and heard some noise in the stairwell. My dad started arguing. Something at his work did not work out; he didn’t reveal what happened. He was a very limited person. He was seeking an occasion to fight. I interrupted him to prevent him from hitting my mother. He ran into the kitchen, and I heard knives clink and my mother started to scream. I was running to help my mother. I was in shock. And I couldn’t accept that my dad took the knife on me. […] I don’t know why I stopped in the stairwell, and he jumped out after me and I wanted to hit him lightly. I hit him and he fell down the stairs.

### 3.3. Vertical and Horizontal Family Related Criminal Career

Family intergenerational (vertical) and sibling (horizontal) delinquency remains one of the most important risk factors in childhood [25] (p. 7), [26] (pp. 143–144). Except for the rare cases where offenders came from “good” middle-class families, in which no one had ever been incarcerated, most of them experienced (about 60%) both intergenerational (vertical) and sibling (horizontal) delinquency. Conflicts with the law, involvement in criminal activity, violence, and drug and alcohol abuse, which characterized the immediate environment of our respondents, were treated as part of the daily routine.

R:Uhu … Tell me, have any of your loved ones, apart from your dad, been punished, for example?

I:Mom is being punished now [for] insurance enforcement [also] all uncles, that is, three uncles.

Furthermore, the biographies of our subjects seemed to confirm a correlation between family size and criminal path [26] (p. 159). It cannot be said that the larger the family, the greater the risk of children’s involvement in crime, but most juvenile and adult offenders came from large families in which parents paid less attention to individual children and did not socialize them to conform to socially desirable norms and values [52,53,54].

R:Do you have siblings?

I:Yes, five brothers and one sister.

R:Did they have any conflict with the law?

I:Later dad died, everybody went into crime, and they got sentences as well. […] only my sister never served time in prison. For example, my brother, Kamil, he was at the age between 13 and 21 in all types of juvenile correctional institutions. [He committed] some robberies, thefts as well.

### 3.4. Family Background and Male Figures

Most of our interviewees painted a picture of a family that had been disrupted in some way. Particularly evident was the large overrepresentation of those who were raised in broken families, which mostly lacked a father figure. Our data showed that the male family figures who appeared in the early life stories analyzed were repeatedly portrayed as anti-models (perpetrators of domestic violence, compulsive alcoholics or drinkers, womanizers, and destroyers of family life). Particularly striking in this context was the tendency for narrators to reproduce the same scenarios in their adult biographies, despite being perceived as destructive. On the basis of psychological opinions (182), we divided biological families of juvenile offenders into functional and dysfunctional ones, taking into account whether the family was broken or not. Table 1 sheds some light on the relationship of family background to future involvement in criminal careers.

The following statistics were confirmed by the experts we questioned. Educators and psychologists working in correctional facilities indicated a lack of or a disturbed male character in the family.

“In my group, 80% of juveniles have family problems. I cannot say categorically that only bad boys who have family problems go to juvenile detention center. There are also boys who have full families, but some neglect appeared in the past. However, 80% of my boys are people who have a gigantic family-related problem. They come from incomplete families where there is no father or stepfather. The father figure itself is very disturbed. Youths are brought up by mothers who cannot cope, either by their grandmother, or they are in an orphanage. Then, there is no male element at all”.(educator, juvenile correctional institution)

The wealth of the family and the possible absence of the father could be seen either as a turning point, freeing the family from the influence of the destructive member or, conversely, as a major contributor to the (further) deterioration of the family’s social and economic situation. This event or situation could have had a direct impact on the economics of the household (loss of an important or main source of income). Alongside this, there were also indirect problems, i.e., the need to reallocate roles in the family to fill the gap left by the absent parent. Some interviewees had to take over the role of chaperone for younger siblings.

I:Well, first they caught me for some theft in a mall when I was a kid. [I stole] some sweets … It was a horrible shame for me. They could have let me go, right? But they called police.

R:And that was your first contact with the police?

I:Yes, when I was a kid. It was a shame. […] And then money was needed, right? I also helped my mother a bit because as I said at the beginning, it was not good. I wanted my brother to have some money. The brothers had to have something to eat, right?

Others entered the workforce prematurely to support the household economically. In each of these cases, interviewees ended up neglecting other responsibilities, primarily dropping out of school or experiencing serious problems in school achievement, which translated into their poorer social capital in the future [55,56]. Criminal activity tends to dominate pro-employment attitudes and becomes a routine form of income generation [57] (p. 64). Our respondents were primarily involved in first-time burglary, theft, and other illegal means of making money and supporting the household budget.

“During school, we went to dig coal illegally in the excavation. We got some cash for one bag. Other colleagues did the same. We had some problems at school and with police. Some of them are in juvenile detention centers or prisons.”(juvenile offender)

As the analyzed biographies show, early criminal initiation caused serious disturbances in the scope and substance of a young person’s contacts with public institutions. Above all, it was often accompanied by dropping out of school or discontinuing education in its early stages [58] (p. 34), [59] (p. 73), [60] (p. 113). Furthermore, our research indicated that low educational attainment is correlated with delinquency and early contact with law enforcement and other institutions set up to police the public, such as youth employment centers, boarding schools, or correctional facilities [54,61].

## 4. Discussion

Risk factors, such as family background and early school drop-out, can have direct or indirect consequences. In both cases, they are seen in a broader time perspective and should be considered in the full context of the course of a person’s biography. For example, a direct consequence of dropping out of school is the absence of an important socialization environment during the individual’s identity-formation stage. With school dropout in the early stages of biography, the level of social control drastically decreases. Indirectly, however, as a person assumes adult roles (during adolescence), he or she is subjected to other pressures, and his or her potential (capital) social capital deteriorates. A poorer education means a worse position in the job market and a lower income than a young person would expect, which can lead to a propensity for criminal activity. Criminologists [58] (p. 34), [59] (p. 73), [62] (pp. 7–24) have indicated that low education is correlated with crime. People in conflict with the law graduate from inferior schools compared to the rest of society, and their educational careers tend to be shorter. Harlow [63], the author of the Education and Correctional Populations report for the Bureau of Justice Statistics, conducted a meta-analysis of five surveys on the professional qualifications of American prisoners. She stated that over 40% of people detained in prisons have only primary education. Bruce Western, a sociologist at Princeton University, estimates that about 70% of inmates do not even have secondary education [64] (p. 9). In his earlier work, based on the National Corrections Reporting Program data from 1983–2001, the author showed that people who did not graduate from high school experienced incarceration five times more often than those with higher education in the analyzed period [59] (p. 73). Low education of convicts is also common in Australia. Baldry [65] (p. 10) found that 75% of the prison population had not graduated from high school.

The strong side of this study was the exceptional amount of original study material from 12 years of our own different research combined and collated with reflections over world-wide literature. A limitation of this research was the lack of longitudinal data analyzed, while life-course criminology is generally considered longitudinal in nature. The other limitation was our focus mainly on the qualitative approach. Using both qualitative and quantitative methodologies (mixed methods) would have enabled us to benefit from both the bigger picture and in-depth insight into interviewees’ narrations, as well. However, using the LCT perspective we benefitted from a theoretical perspective that helped us to frame and explore some of the risk factors that we noted during the inductive analysis and reasoning based on qualitative data. For further attempts to study risk factors in offenders’ group, we see a need for a slightly different methodological approach. It is possible that going for an interdisciplinary methods model (psychological, sociological, biological) would allow us to clear some gaps in modern criminology. It seems that this might be another path for exploring the risk factors of a criminal career. It is only one of a dozen to help researchers understand the studied problem, in order to be able to propose better preventive individualized systems, like risk-factor interventions. There have already been some attempts to initiate and facilitate the debate concerning the interaction between factors belonging to a variety of different scientific fields and to provide an interdisciplinary approach to the study of criminal behavior etiology [66]. For instance, there has been a growing number of publications concerning the biological consequences of psychological experiences and disorders, including traumatic events. Among the outcomes of those studying children are problems with developing resilience (due to damages in biological stress systems), cognitive functioning, and brain development [67]. Other studies have shown implications of biological factors on criminal behavior. In 2018, Williams [68] published an article suggesting that traumatic brain injury may be one of the causes not only for developing a criminal career, but also for continuing it. It may procure agents like poor engagement in treatment, in-custody infractions, and reconviction.

Some LCT-oriented researchers have also begun to explore the physiological and neurological roots associated with problem behaviors and desistance [69,70,71]. They revealed that during adolescence, there is a “sharp increase in dopaminergic activity in the limbic and paralimbic areas of the brain, characterized as the socioemotional system, which leads to increases in reward seeking and risk taking in adolescence” [69] (p. 789). Biology-related factors shed new light on the benefits of an interdisciplinary approach to criminology [66,72].

## 5. Conclusions

Our data revealed that, with increasing age and autonomy, the subjects were influenced by peers and, to a lesser extent, by relatives and neighbors. It was the acquaintances who accompanied our respondents in taking their first steps on the criminal path, which often occurred during escapes from home. The narratives of our subjects demonstrated that the onset of delinquency should be considered as different age-related periods, rather than just as a general onset, since different risk factors are present at different ages.

In contrast to adolescents, adults are more negative about the role of fathers in their upbringing. In most cases, fathers were responsible for transmission of criminal values and a lack of respect to law and social norms. For those fathers who did not engage in domestic violence but were aggressive outside the family, juvenile offenders tended to glorify their fathers as “tough guys”. Both juvenile and adult offenders evaluated fathers involved in domestic violence negatively and did not maintain contact with them.

Intergenerational (vertical) and sibling (horizontal) delinquency that our subjects experienced in early childhood had a great impact both on their crime onset and criminal career involvement. Our in-depth interviews confirmed that offenders came from families in which relatives had been incarcerated. The majority of our interviewees experienced both intergenerational (vertical) and sibling (horizontal) delinquency. Early-childhood criminal onset was also associated with poor school achievements and early drug/alcohol initiation. Furthermore, we noticed that family size matters in the criminal career. Most juvenile and adult offenders came from large families, in which parents paid less attention to individual children and did not socialize them to conform to socially desirable norms and values.

## Figures and Tables

**Table 1 ijerph-18-06583-t001:** Family background of juvenile offenders located in correctional institutions.

Type of Family	Dysfunctional Family (%)	Family without Signs of Dysfunction (%)	Total
Two-parent family (including reconstructed)	43 (23.6)	43 (23.6)	86 (47.2)
Single-parent family	71 (39)	25 (13.7)	96 (52.7)
Total	114 (62.6)	68 (37.4)	182 (100)

Source: Chomczyński 2017: 225 [19].

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
