# Peer review of "Risk Factors of Early Adolescence in the Criminal Career of Polish Offenders in the Light of Life Course Theory"

_ijerph, 2021, doi:10.3390/ijerph18126583_

Round 1

Reviewer 1 Report

Dear authors,

First of all I would like to congratulate you for such an interesting piece of research. The topic is not just appealing but also quite up-to-date . The design of the research is appropiate and so the methods are. As a minor revision I would suggest you improve your conclusions as they are a little shorter than expected taking into consideration the engaging results of the research.

Author Response

Dear Sir/Madam,

        We wish to express our appreciation for the comments and suggestions for our manuscript entitled “Risk factors of early adolescence in the criminal career of Polish offenders in the light of life course theory”. We have carefully revised the manuscript taking into consideration all the comments.

  • First of all I would like to congratulate you for such an interesting piece of research. The topic is not just appealing but also quite up-to-date . The design of the research is appropriate and so the methods are.

Thank you very much.

  • As a minor revision I would suggest you improve your conclusions as they are a little shorter than expected taking into consideration the engaging results of the research.

We have made the conclusions more detailed.

Once again we would like to sincerely thank you for a very comprehensive, insightful and crediting review. All of the above comments are highly valuable for the comprehensiveness of the paper and our own scientific development.

Kind Regards,

Dariusz Timler Assoc. Prof., PhD, MD

Reviewer 2 Report

Throughout the paper the authors refer to individuals they cite with their first and last name.  I would suggest changing this to just last names. 

Pg. 2 line 65 – the authors suggest that juvenile delinquency is of particular interest to criminology.  While this is true, I would suggest framing this as a particular interest to life-course criminology as that is what you are focused on. 

Pg. 2 line 70 – authors state that through the years many studies have addressed the topic but then fail to cite them.   

Pg. 2 line 76 -  While it is true that the family can act as a protective factors that will insulate an individual from criminogenic behavior, the inverse is also true.  The authors did a good job discussion this.  However, some mention of Adverse Childhood Experiences (ACEs) and citations would help bolster this area. 

Pg. 5 line 223 – is “even often.” A typo?

Pg. 10 line 413 – need a space between with and problem

One of the major limitations to this research is the lack of longitudinal data analyzed.  Life-course criminology is generally thought to be longitudinal in nature (i.e. Cambridge Study in Delinquent Development).  It is my understanding that really only the third project mentioned has true life-course data.  This should at least be mentioned in the paper.

I suggest the authors justify the use of life-course theory in the back end of the paper.  This would create a more cohesive analysis.  

Author Response

Dear Sir/Madam,

        We wish to express our appreciation for the comments and suggestions for our manuscript entitled “Risk factors of early adolescence in the criminal career of Polish offenders in the light of life course theory”. We have carefully revised the manuscript taking into consideration all the comments.

  • Throughout the paper the authors refer to individuals they cite with their first and last name.  I would suggest changing this to just last names. 

We changed the citations to just last names of the authors.

  • 2 line 65 – the authors suggest that juvenile delinquency is of particular interest to criminology.  While this is true, I would suggest framing this as a particular interest to life-course criminology as that is what you are focused on. 

We specified that juvenile delinquency is a particular interest to life-course criminology. It is visible in the introduction.

  • 2 line 70 – authors state that through the years many studies have addressed the topic but then fail to cite them.   

We added the citations for our statement.  

  • 5 line 223 – is “even often.” A typo?

We corrected this mistake.

  • 10 line 413 – need a space between with and problem

We corrected this mistake.

  • One of the major limitations to this research is the lack of longitudinal data analyzed.  Life-course criminology is generally thought to be longitudinal in nature (i.e. Cambridge Study in Delinquent Development).  It is my understanding that really only the third project mentioned has true life-course data.  This should at least be mentioned in the paper.

We mentioned that the weak point of this research is the lack of longitudinal study that among others is at the center of LCT interest.

  • I suggest the authors justify the use of life-course theory in the back end of the paper.  This would create a more cohesive analysis.  

We added the justification for the use of life-course theory in this paper. It is visible in the discussion.

Once again, we would like to sincerely thank you for a very comprehensive, insightful and in many places accurate review. All of the comments are highly valuable for the comprehensiveness of the paper and our own scientific development.

Kind Regards,

Dariusz Timler Assoc. Prof., PhD, MD

Reviewer 3 Report

Review. ijerph-1233109. “Risk factors of early adolescence in the criminal career of Polish offenders in the light of life course theory”. The paper investigates environmental related risk factors which contribute to the initiation of early adolescence’s criminal behavior. The paper has a clear objective, the research methods and materials are appropriate. The results and conclusions can largely meet the objective. The paper is suitable to IJERPH. I suggest a minor revision before publishing. I list my comments below:

  1. Too much dialogue was put into the paper. More specifically, p5-7, and p9.
  2. I wonder if there’s any gender difference in family related crime risk (section 3.2).
  3. Although the paper used open-ended interviews to collect materials and used qualitative methods to analyze the materials. Given that there are 130 participants, which is enough for quantitative analysis as well, I assume simple quantitative techniques such as 2-way contingency table analysis with chi-squared test can be used to study the link between gender (categorical variable) and certain type of risk factors (categorical variable). This is helpful to reveals the link more accurately.
  4. I suggest adding the x2 test and expected values for each column in Table 1.

Minor comments

  1. Section headings are numbered repeatedly. Line 276 is “3.3”, while line 307 is also “3.3”.
  2. P4, line 145, missing blank space between “to” and “2015”. This issue exists in several other places, such as line 445, 449, 451.

Author Response

Dear Sir/Madam,

        We wish to express our appreciation for the comments and suggestions for our manuscript entitled “Risk factors of early adolescence in the criminal career of Polish offenders in the light of life course theory”. We have carefully revised the manuscript taking into consideration all the comments.

  1. Too much dialogue was put into the paper. More specifically, p5-7, and p9.

We rearranged the dialogue part of this paper and made it more dense.

  1. I wonder if there’s any gender difference in family related crime risk (section 3.2).

The aim of this paper was to elaborate on early adolescence risk factors for building a base for possible prevention. In this paper we try to discuss the inconsistencies with LCT that we observed in our three empirical research on the criminal careers of Polish offenders rather than focus on gender differences. It is an important topic which we would not like to signal in a vague manner. In our future research we plan to focus on sex differences and collect more standardized and comparable data from both women and men.

  1. Although the paper used open-ended interviews to collect materials and used qualitative methods to analyze the materials. Given that there are 130 participants, which is enough for quantitative analysis as well, I assume simple quantitative techniques such as 2-way contingency table analysis with chi-squared test can be used to study the link between gender (categorical variable) and certain type of risk factors (categorical variable). This is helpful to reveal the link more accurately.

In qualitative study like ours it is unlikely to make a separate qualitative analysis. It is a great suggestion for our future work, for instance to use more data and apply quantitative methods (i.e., text mining) to explore the patterns. We plan to apply a quantitative tool (questionnaire) which let us to use more sophisticated statistics (logical regression, factor analysis, correlations). Due to the fact that all 3 projects discussed in this paper were strictly qualitative, we strongly feel that quantitative techniques would not be justified when used to narrations of our interviewees.

  1. I suggest adding the x2 test and expected values for each column in Table 1.

Applying a qualitative approach, we focused on delivering in-depth and exploratory insight in risk factors of early adolescence in criminal careers. Using tools typical for qualitative analysis our aim was to generate categories inductively and serendipitously to understand nuances lying under choices made by offenders. Despite statistical representatives we employed theoretical saturation and theoretical sampling to develop better understanding of family, environmental and peers’ influence on criminal onset and persistence.

Minor comments

  1. Section headings are numbered repeatedly. Line 276 is “3.3”, while line 307 is also “3.3”.

We corrected this mistake.

  1. P4, line 145, missing blank space between “to” and “2015”. This issue exists in several other places, such as line 445, 449, 451.

We corrected those mistakes.

Once again, we would like to sincerely thank you for a very comprehensive, insightful and in many places accurate review. All of the comments are highly valuable for the comprehensiveness of the paper and our own scientific development.

Kind regards,

Dariusz Timler Assoc. Prof., PhD, MD